# APR-246 increases tumor antigenicity independent of p53

Judith Michels[1,2,*] , Divya Venkatesh[1,2,*] , Cailian Liu[1,2], Sadna Budhu[1,2], Hong Zhong[1,2], Mariam M George[1,2], Daniel Thach[1,2] , Zhong-Ke Yao[3] , Ouathek Ouerfelli[3] , Hengrui Liu[4] , Brent R Stockwell[4,5], Luis Felipe Campesato[1,2], Dmitriy Zamarin[6,7,8], Roberta Zappasodi[9] , Jedd D Wolchok[1,2,9],† , Taha Merghoub[1,2,9],†

We previously reported that activation of p53 by APR-246 reprograms tumor-associated macrophages to overcome immune checkpoint blockade resistance. Here, we demonstrate that APR-246 and its active moiety, methylene quinuclidinone (MQ) can enhance the immunogenicity of tumor cells directly. MQ treatment of murine B16F10 melanoma cells promoted activation of melanoma-specific CD8[+] T cells and increased the efficacy of a tumor cell vaccine using MQ-treated cells even when the B16F10 cells lacked p53. We then designed a novel combination of APR-246 with the TLR-4 agonist, monophosphoryl lipid A, and a CD40 agonist to further enhance these immunogenic effects and demonstrated a significant antitumor response. We propose that the immunogenic effect of MQ can be linked to its thiol-reactive alkylating ability as we observed similar immunogenic effects with the broad-spectrum cysteine-reactive compound, iodoacetamide. Our results thus indicate that combination of APR-246 with immunomodulatory agents may elicit effective antitumor immune response irrespective of the tumor's p53 mutation status.

## Introduction

Immune checkpoint blockade (ICB) has led to considerably improved prognosis of cancer patients across several different tumor types (i.e., melanoma, clear cell renal, non-small cell lung, head and neck, cervical, endometrial cancer, Hodgkin lymphoma), either as a monotherapy or in combination with other immunotherapies, chemotherapy or targeted therapies (Reck et al, 2016; Gandhi et al, 2018; Burtness et al, 2019; Larkin et al, 2019; Motzer et al, 2019; Powles et al, 2020; Colombo et al, 2021; Kuruvilla et al, 2021; Makker et al, 2022). However, only a subset of patients experience significant long-term benefit with durable responses and there is a need for better predictive biomarkers (Bruni et al, 2020; McGrail et al, 2021; Fridman et al, 2022). Currently, there are a few main strategies being developed to enhance tumor immune infiltration and extend the antitumor immune response in patients. Either passively, by supplementing the patient's T cell repertoire with the immune effector cells (e.g., the adoptive transfer of tumor-infiltrating lymphocytes or chimeric antigen receptor T-cell therapy) (Stevanović et al, 2015; Schuster et al, 2019), or actively by modulation of the tumor microenvironment (TME) (e.g., intratumoral delivery of oncolytic viruses [Ramelyte et al, 2021], or by other therapies such as radiotherapy, photodynamic therapy [O'Shaughnessy et al, 2018], messenger RNA-stabilized lipid-based nanoparticles [NCT03897881] [Sahin et al, 2020], T cell engagers [Goebeler & Bargou, 2020], and anti-angiogenic agents [Fukumura et al, 2018]) for instance.

APR-246/PRIMA-1[MET] (eprenetapopt, Aprea Therapeutics) is a novel small molecule anti-cancerous compound that stabilizes p53 in the active conformation by conjugation to thiol groups (AKA sulfhydryl [SH] groups) (Lambert et al, 2009; Degtjarik et al, 2021). APR-246 is a synthetic prodrug, which spontaneously hydrolyzes to the active drug 2-methylene-quinuclidin-3-one (MQ), a Michael acceptor that alkylates the thiol group of cysteines on proteins such as p53.

We previously reported enhanced antitumor activity when APR-246 was combined with either anti-PD-1 alone or dual blockade of PD-1 and cytotoxic T-lymphocyte antigen 4 protein (CTLA-4) in multiple murine tumor types (Ghosh et al, 2022). Our data suggested that APR-246-mediated reprograming of tumor-associated macrophages in the TME augments the response to ICB in a p53-dependent manner. Based on this study, the combination of APR-246 and the anti-PD-1 antibody pembrolizumab has been investigated in a clinical trial in solid tumors (i.e., ICB-naïve gastric and bladder cancer and non-small cell lung cancer patients progressing on prior anti-PD-1/L1 therapy) with a 13.8% clinical benefit rate (Park et al, 2022). Whereas this prior preclinical work focused on dissecting the effect of p53 stabilization in immune cells, we also

[1]Department of Pharmacology, Swim Across America and Ludwig Collaborative Laboratory, Weill Cornell Medicine, New York, NY, USA [2]Sandra and Edward Meyer Cancer Center, Weill Cornell Medicine, New York, NY, USA [3]The Organic Synthesis Core Facility, MSK, New York, NY, USA [4]Department of Biological Sciences, Columbia University, New York, NY, USA [5]Department of Chemistry, Columbia University, New York, NY, USA [6]Swim Across America and Ludwig Collaborative Laboratory, Immunology Program, Parker Institute for Cancer Immunotherapy, Memorial Sloan Kettering Cancer Center, New York, NY, USA [7]Immuno-Oncology Service, Human Oncology and Pathogenesis Program, Memorial Sloan Kettering Cancer Center, New York, NY, USA [8]Department of Medicine, Memorial Sloan Kettering Cancer Center, New York, NY, USA [9]Department of Medicine, Weill Cornell, New York, NY, USA

Correspondence: tmerghoub@med.cornell.edu; jwolchok@med.cornell.edu
*Judith Michels and Divya Venkatesh contributed equally to this work
†Jedd D Wolchok and Taha Merghoub jointly supervised this work

observed an increase in APCs and CD8[+] T cells in the TME of patients and mice receiving APR-246 treatment. These effects led us to postulate that APR-246 may directly enhance the immunogenicity of tumor cells themselves. Thus, in this study, we wanted to test if APR-246 or its active moiety methylene quinuclidinone (MQ) can directly increase the immunogenicity of tumor cells independent of its effects on the immune cells of the host.

It is also well known that APR-246 triggers cellular stress responses (i.e., endoplasmic reticulum and oxidative stress) in a p53-independent manner by interfering with antioxidant pathways through the formation of protein adducts pursuant to modification of cysteines and selenocysteines of glutathione and thioredoxin reductase, respectively (Peng et al, 2013; Mohell et al, 2015; Liu et al, 2017; Haffo et al, 2018; Milne et al, 2021). Because cellular stress response pathways may have profound immunomodulatory functions in tumor cells (Ishimoto et al, 2011; Michaud et al, 2011; Senovilla et al, 2012; Duan et al, 2019; Li et al, 2019; Fox et al, 2020; Chen & Cubillos-Ruiz, 2021; Guo et al, 2021), we also wanted to evaluate whether APR-246 can impact the antigenicity of tumor cells even in the absence of p53.

In this study, we show that MQ-treated tumor cells enhanced the activation and proliferation of tumor-specific CD8[+] T cells with long-lasting antitumor immunity when used in the form of a tumor cell vaccine. APR-246 combined with the TLR-4 agonist mono-phosphoryl lipid A (MPLA) further enhanced tumor infiltration with APCs and activated CD8[+] T cells. The addition of a CD40 agonist to this combination was able to significantly reduce melanoma progression. Importantly, these effects, albeit reduced, were still present in B16F10 p53 null tumors. Thus, we hypothesize that the immunogenic effects of MQ may be associated with its ability to bind cysteines of proteins other than p53. Overall, this study strengthens the rationale of combining APR-246 with immune modulators in patients irrespective of tumor's p53 status. Moreover, it suggests that other drugs that are able to covalently attach to thiol groups within tumors may also promote tumor immunogenicity and can thus constitute rational partners for novel combinations with immunotherapy to induce long-lasting antitumor immunity.

# Results

## APR-246/MQ-treated tumor cells induce specific T-cell responses in vivo irrespective of the tumor's p53 status

To evaluate the impact of MQ (the APR-246 active moiety) treatment on the immunogenicity of tumor cells, we used GVAX as an immunization strategy. It consists of inoculating irradiated tumor cells engineered to secrete GM-CSF, to promote antitumor immunity (Dranoff et al, 1993; van Elsas et al, 1999). We modified this GVAX protocol to include B16F10-GM-CSF melanoma cells treated with MQ to test the efficacy of priming specific T cell responses in vivo (Fig 1A). We tested either a short-term pulse treatment (10 $\mu$M for 4 h, followed by culture in fresh media for 48 h) or continuous treatment (10 $\mu$M for 48 h) with MQ in vitro (Fig S1A and B). The pulsed MQ regimen was less cytotoxic to B16-GMCSF cells than continuous dosing (Fig S1A). Moreover, continuous treatment showed a

decrease in MHC-I (H2-K[b]) and MHC-II in treated B16 tumor cells upon IFN-$\gamma$ exposure (Fig S1B). We have also previously reported that pulsatile treatment regimens can be better suited for small molecules that elicit an immunogenic response in the tumor (Choi et al, 2019).

T cell responses against melanocyte-specific antigens such as melanosomal proteins involved in pigment synthesis (i.e., gp100, tyrosinase, MART-1/Melan-A, TRP1, and TRP2) correlate with tumor regressions and vitiligo in vivo in mice (van Elsas et al, 2001; Overwijk et al, 2003) and patients with melanoma (Rosenberg & White, 1996; Hua et al, 2016). B16F10 (hereafter B16) murine melanoma cells express gp100, which contains immunodominant epitopes that can be targeted by tumor infiltrating lymphocytes in both mice and humans (Kawakami et al, 2000; Cohen et al, 2006; Yuan et al, 2009). Because adoptive transfer of naïve antigen-specific CD8[+] T cells increases vaccine-elicited tumor immunity (Overwijk et al, 1998, 2003; Rizzuto et al, 2009), we coupled the MQ-treated tumor cell vaccine with adoptive transfer of naïve Pmel-1 TCR-transgenic gp100-specific CD8[+] T cells (pmel) (Fig 1A). This strategy allowed us to precisely monitor activation and proliferation of these tumor antigen-specific CD8[+] T cells as means to evaluate the in vivo effects of potential increases in antigenicity of tumor cells post MQ treatment. We inoculated vehicle- or MQ pulse-treated irradiated B16 GM-CSF tumor cells into syngeneic C57BL/6 mice that previously received adoptively transferred naïve CFSE-labeled Thy1.1 congenically marked, antigen-specific pmel CD8[+] T cells (Fig 1A). The irradiated MQ-treated B16-GMCSF cells used for vaccination were non-apoptotic (Fig S1C) as measured by the simultaneous cytofluorometric assessment of mitochondrial transmembrane potential ($\Delta\psi$m) and plasma membrane permeabilization after staining with the $\Delta\psi$m-sensitive fluorochrome 3,3′-dihexyloxacarbocyanine iodide [DiOC6(3)] and the exclusion dye propidium iodide (PI) (Kepp et al, 2011). And the MQ treatment did not significantly impact the GMCSF expression of B16-GMCSF tumor cells (Fig S1D). 6 d later, we analyzed the proliferation (CFSE dilution) of the transferred (Thy1.1[+]) pmel CD8[+] T cells harvested from tumor-draining lymph nodes to assess priming and antigen-specific response. Using the above approach, we observed enhanced proliferation of pmel CD8[+] T cells (compared with vehicle-treated cells), as reflected by an increase in the CFSE dilution (CFSE[low]) in the draining lymph nodes of MQ pre-treated tumors (Fig 1B), suggesting that MQ treatment can increase tumor immunogenicity.

In parallel, we also observed a significant increase in MHC-I (H2-K[b]) and MHC-II expression on B16 tumor cells that were treated with MQ in vitro (Fig 1C) and an increase in MHC-I (H2-K[b]) expression with an equivalent pulse treatment with APR-246 in vivo (bid, twice a week) (Fig 1D). These data suggest that treatment with MQ in vitro or APR-246 in vivo leads to increased tumor antigen presentation that can enhance the priming of tumor-specific T cell responses.

MQ is known to reactivate transcriptional function of unfolded WT or mutant p53 regardless of the mechanism causing its deactivation (Lambert et al, 2009; Degtjarik et al, 2021). To evaluate if there is a link between p53 activation and the measured immunogenic effect in the tumor cells upon MQ treatment, we knocked out Trp53 (Trp53[−/−]) in B16 cells using the CRISPR/Cas-9 system (Fig S1E and F). As expected, there was no p53 protein expression in Trp53[−/−] B16 cells when compared with control B16 (WT) cells that

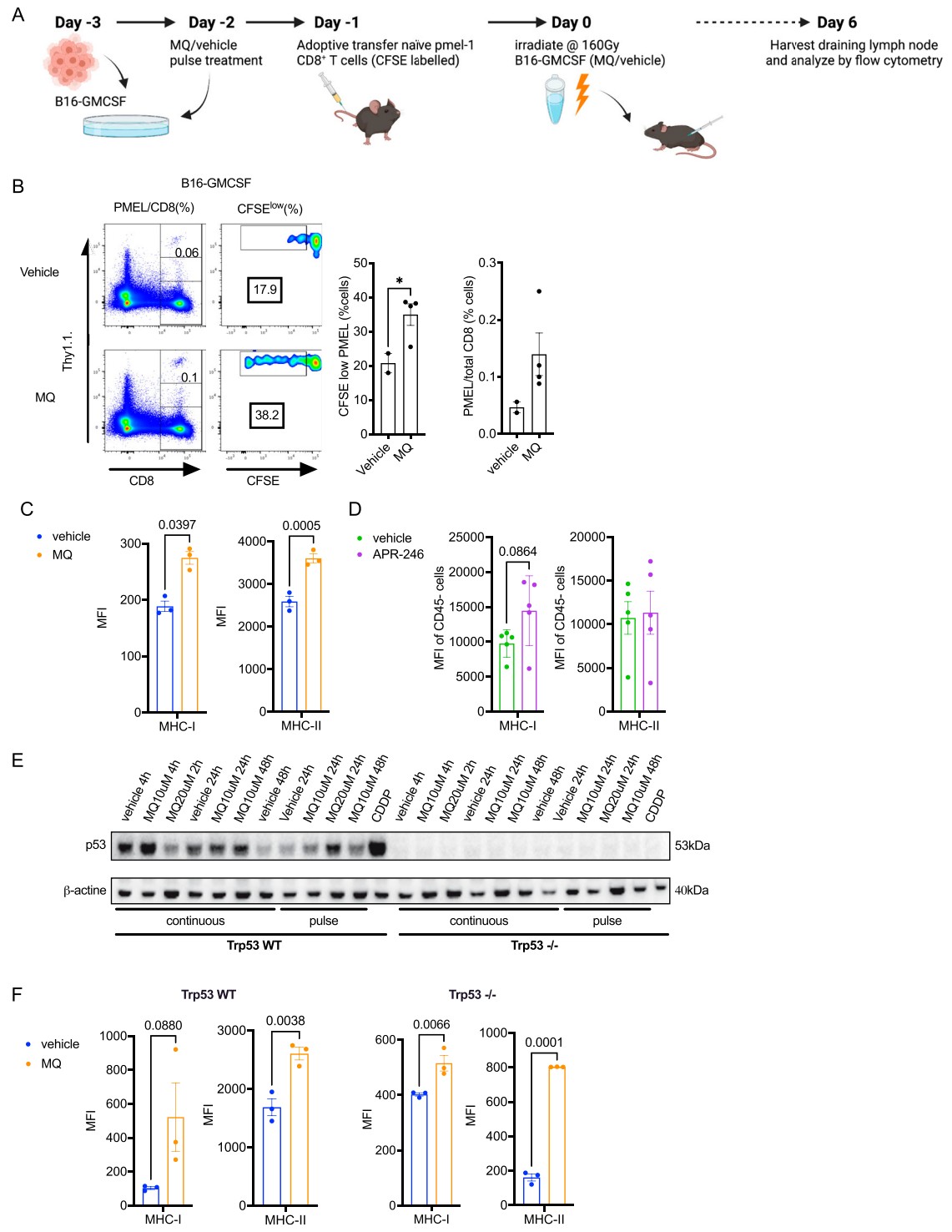

**Figure 1. MQ-treated B16-GMCSF cells increase the proliferation of antigen-specific pmel CD8⁺ T cells in vivo.**
**(A)** The schematic for the vaccination of C57BL/6 mice with B16-GMCSF cells that had undergone an in vitro treatment with MQ and adoptive transfer of Pmel CD8⁺ T cells. **(A, B)** Left panel—representative flow images for CFSE dilution of pmel CD8⁺ T cells harvested from the draining lymph node of C57BL/6 mice implanted with B16-GMCSF cells according to the schematic in (A). **(B)** Middle panel—fraction of CFSE^low (proliferating) pmel CD8⁺ T cells among total CFSE⁺ pmel CD8⁺ T cells. **(B)** Right panel—proportion of pmel CD8⁺ T cells among total CD8⁺ cells in the lymph node. **(C)** Mean fluorescence intensity (MFI) (MFI of antibody/MFI of isotype control) of the MHC class I (H2Kb) and the MHC class II (IA/IE) as evaluated by flow cytometry in (C). **(D)** B16 cells that were treated in vitro with MQ (10 µM)/vehicle for 4 h and harvested 24 h later or (D) in B16 tumors from mice treated with APR-246 (pulse treatment of 100 mg/kg 2qd biweekly x2). **(E)** Effect of in vitro MQ treatment on the Trp53 pathway in B16-derived WT and Trp53 CRISPR/Cas9 knock-out (Trp53⁻/⁻) cells as measured by Western blot. The treatment was done either in a continuous or pulse fashion (pulse treatment with MQ/vehicle for 2–4 h, drug removal, and cell culture in fresh media for 24/48 h). CDDP, cisplatin (40 µM) treatment for 24 h was used as a positive control for p53 activation. β-Actin was used as a loading control. **(F)** The MFI of the MHC class I (H2Kb) and the MHC class II (IA/IE) as evaluated by flow cytometry in B16 WT or Trp53⁻/⁻

underwent a similar editing process using the CRISPR/Cas-9 system but with a control untargeted vector instead. In Trp53$^{-/-}$ cells, as expected, neither MQ continuous nor pulsed treatment was able to induce Trp53 expression similar to treatment with cisplatin (CDDP) that was used as a positive control for Trp53 induction (Fig 1E). Nevertheless, despite the lack of p53 expression, we continued to observe a significant increase in MHC surface expression in Trp53$^{-/-}$ cells upon pulse treatment with MQ as compared with vehicle control (Fig 1F). In addition, we found similarly enhanced proliferation (i.e., cell trace violet [CTV] dilution) and activation (i.e., CD44 expression) of pmel CD8$^+$ T cells that were adoptively transferred in mice which were immunized with irradiated MQ-treated WT or Trp53−/− B16 cells (even in the absence of GMCSF) (Fig S2A–E). This indicates that the measured immunogenic effect of MQ treatment in tumor cells is not entirely dependent on activation of the p53 protein.

## Long-term immunizing effects of MQ-treated tumor cell vaccine

Because our MQ-treated GVAX elicited an enhanced proliferation of adoptively transferred tumor antigen-specific T cells (Fig 1), we then tested the extent to which MQ treatment could enhance the therapeutic antitumor efficacy of the vaccine as the next logical step. GVAX/MQ or GVAX/vehicle was administered three times starting on the day of tumor implantation (Fig 2A). We observed a trend of tumor growth reduction and prolonged survival of mice immunized with GVAX/MQ compared with mice immunized with GVAX/vehicle (Fig 2B). We also detected significantly enhanced endogenous systemic tumor antigen-specific CD8$^+$ T cell response in GVAX/MQ as compared with GVAX/vehicle immunized mice. This was measured by the amount and activation (i.e., CD44$^+$CD62L$^-$ and PD-1$^+$ fraction) of tetramer-positive gp100$_{25-33}$/D$^b$ reactive (and negative control SIINFEKL-specific) CD8$^+$ T cells in the blood of mice, a week after the end of the vaccination cycle (Fig 2C and D). As expected, the impact of GVAX vaccination was lost in mice that lacked dendritic cells (Batf3$^{-/-}$ mice) or mature T and B cells (Rag$^{-/-}$) (Fig S2F–I). Finally, we assessed the impact of p53 expression in the tumor cells on the therapeutic effect of GVAX/MQ. We prepared a GVAX using a combination of B16 Trp53$^{-/-}$ cells treated with MQ/vehicle and a small fraction (i.e., B16 Trp53$^{-/-}$: B16-GMCSF cell ratio of 5:1) of untreated B16-GMCSF cells (as a source for the GMCSF cytokine) (Fig 2E). The trend for an improved antitumor therapeutic effect existed even in these cells although to a lesser degree, thus supporting the conclusion that the immunogenic effect of the APR-246 is not necessarily dependent on p53 expression in the tumor.

Because the overall impact of GVAX on survival was moderate (Fig 2B), we combined GVAX with CTLA-4 blockade, which is known to enhance T-cell priming and improve the effect of immunization (van Elsas et al, 1999). CTLA-4 blockade further delayed tumor growth independent of MQ pretreatment for the tumor cells in the vaccine (Fig 3A). However, the use of GVAX/MQ in combination with anti-CTLA-4 achieved the best overall survival results, with ~80%

tumor-free mice at 100 d post treatment (Fig 3B–I). Furthermore, all cured mice in this group completely rejected a secondary tumor reimplantation 120 d later (Fig 3B). This result further underscores the relevance of MQ treatment to prime long-lasting antitumor T cell responses.

## MQ directly enhances antigen presentation in treated tumor cells

To further characterize the direct effects of MQ on tumor cells, we developed an in vitro co-culture assay and assessed the proliferation and activation of pmel CD8$^+$ T cells when exposed to irradiated MQ-treated B16 tumor cells (Figs 4A and S3A). Of note, the pmel CD8$^+$ T cells are not exposed to the MQ drug in this co-culture system and only the tumor cells are treated. In line with our previous results, MQ-treated B16 cells were able to directly prime pmel CD8$^+$ T cells as measured by increased proliferation and expression of T cell activation markers (left panels of Figs 4B and C and S3C). The increase in proliferation of these T cells is represented as multiple generations based on the degree of dilution of the fluorescent CTV dye, where the generation number is indicative of the number of cell divisions that the T cells underwent (Fig S3B left panel). When B16 Trp53$^{-/-}$ cells were used in this assay, we found that the effect on T-cell activation was maintained albeit reduced (Figs 4B and C and S3B right panels). This supports the conclusion that the increased tumor immunogenicity because of MQ treatment is at least partly independent of p53. However, differentiating between the direct effects on the antigen presentation machinery of the tumor cells versus other indirect effects on the cells is limited with this system. Thus, we modified the in vitro co-culture assay by first exposing BMDCs CD11c$^+$ to the irradiated MQ/vehicle-treated tumor cells, and then incubating the naïve T cells (pmel CD8$^+$) with these tumor-exposed CD11c$^+$ dendritic cells (Fig 4D). Note that in this setting, neither the BMDCs nor the T cells are directly exposed to MQ. The exposure to tumor cells increased the expression of activation markers on the BMDCs as expected, and the treatment of tumor cells with MQ significantly boosted this activation of BMDCs (Fig 4E). Further validating the direct effect of APR-246 treatment on tumor antigenicity, BMDCs loaded with MQ-versus vehicle-treated B16 tumor cells enhanced the priming of cocultured pmel CD8$^+$ T cells by increasing their proliferation and activation (Figs 4F and G and S3D and E left panels). As described previously, the increase in proliferation of these T cells is represented as multiple generations based on the degree of dilution of the fluorescent CTV dye. Interestingly, the BMDCs exposed to MQ-treated B16 Trp53$^{-/-}$ cells were also more activated and were able to enhance the priming and activation of pmel CD8$^+$ T cells (Figs 4E–G and S3D and E right panels), although the degree of activation of the T cells was reduced when compared with the B16 WT cells. The results of both co-culture assays and the immunization strategy using MQ-treated tumor cells strongly support tumor-specific immunogenic effects of MQ, which may occur to different degrees depending on p53 expression in the tumor.

cells when treated in vitro with MQ (10 μM)/vehicle for 4 h and harvested 24 h later. The data represent mean ± SEM and the P-value was calculated by unpaired parametric t test for the in vivo data and two-way ANOVA test for the in vitro data, *<0.0332, **<0.0021, ***<0.0002, ****<0.0001. n = 3 for in vitro experiments or n = 5 for in vivo experiments and is representative of three independent experiments.

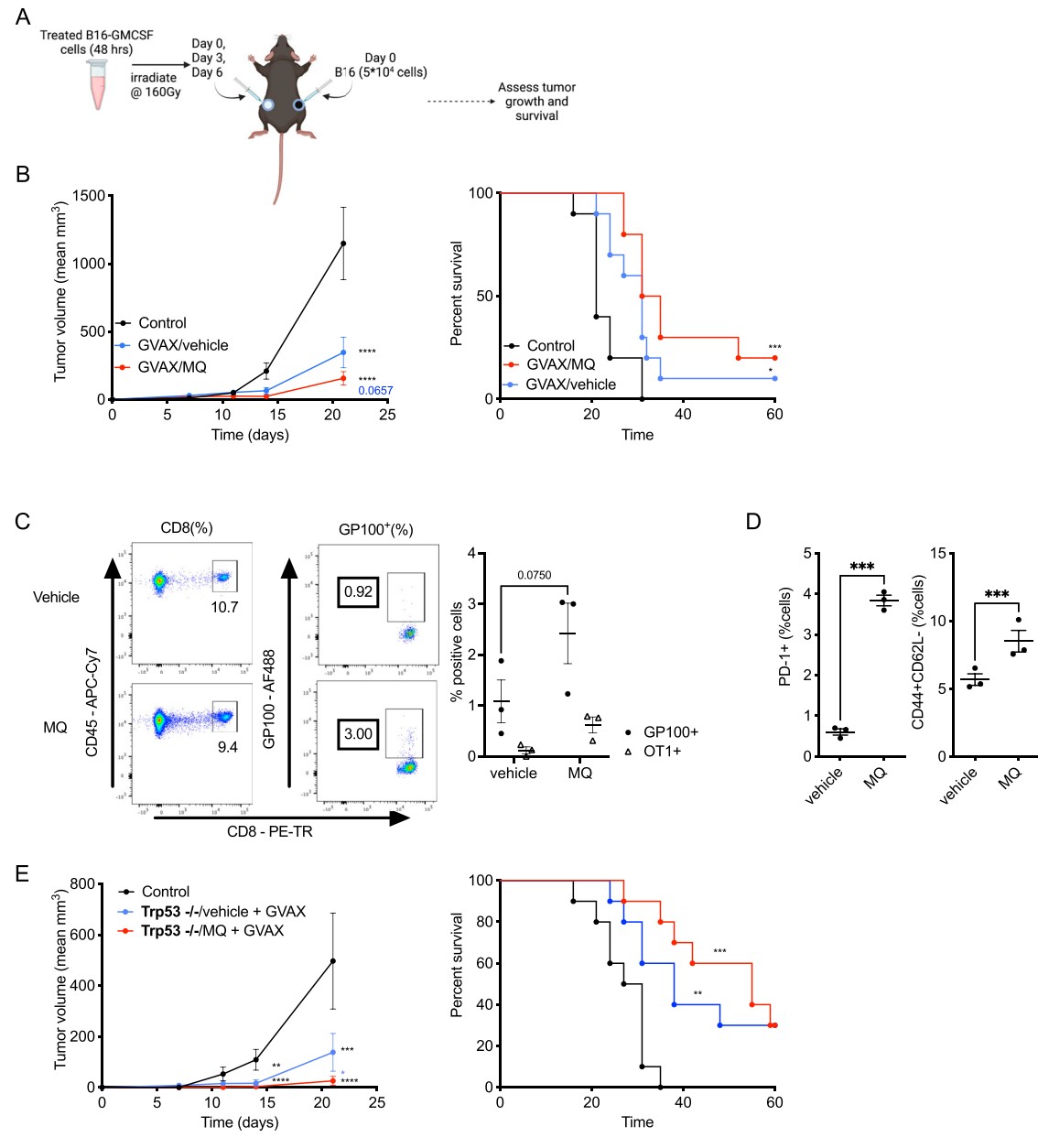

**Figure 2. Tumor cell immunization using GM-CSF–producing B16 tumor cells (GVAX) treated with MQ improves tumor rejection.**
**(A)** The schematic of the corresponding GVAX treatment. The B16-GMCSF cells were treated with MQ (10 $\mu$M)/vehicle for 4 h and then cultured for 48 h in fresh media.
**(B)** Mean tumor size progression overtime (left panel) and survival (right panel) of nonimmunized (control) and immunized (GVAX/vehicle or MQ) mice. **(C)** Peripheral blood was collected at D13 after first dose of immunization (i.e., 7 d after the third dose) to assess the presence of live CD8$^+$ gp100$_{25-33}$/D$^b$ tetramer-positive T cells. SIINFEKL peptide was used as a negative control (n = 3). Representative flow images (left panel) and quantification of the tetramer reactive cells (right panel). **(D)** Activation markers (i.e., PD-1$^+$ and CD44$^+$CD62L$^-$) among the GP100$_{25-33}$/D$^b$ tetramer-positive CD8$^+$ T cells. **(E)** Mean tumor size progression overtime and survival in mice immunized with a mixture of B16 Trp53$^{-/-}$ vehicle/MQ-treated and B16-GMCSF non-treated cells. The cells were mixed before irradiation and before injection in mice. The data represent mean ± SEM and the P-value is represented as *<0.0332, **<0.0021, ***<0.0002, ****<0.0001. P-value was calculated by two-way ANOVA for flow panels and tumor growth curves and by log-rank (Mantel–Cox) test for survival curves. n = 10 and representative of two (C, D) to three experiments.

## Improved antitumor effects of APR-246 treatment in combination with TLR4 and CD40 agonists

We reasoned that the potential increase in tumor immunogenicity because of MQ could be better translated in vivo by further activating APCs. To this end, we combined APR-246 with a TLR4 agonist, which can aid in the optimal maturation of APCs (Khalil et al, 2019).

MPLA, a TLR4 agonist, is reported to activate dendritic cells and favor antigen presentation for the initiation of adaptive immune responses (Ismaili et al, 2002; Didierlaurent et al, 2009). We evaluated if intratumoral MPLA could further potentiate the antitumor immunogenic effects of APR-246 in vivo. We treated C57BL/6 mice implanted with B16 tumor cells using a combination of MPLA with either APR-246 dosed as (i) a continuous treatment of 100 mg/kg i.p

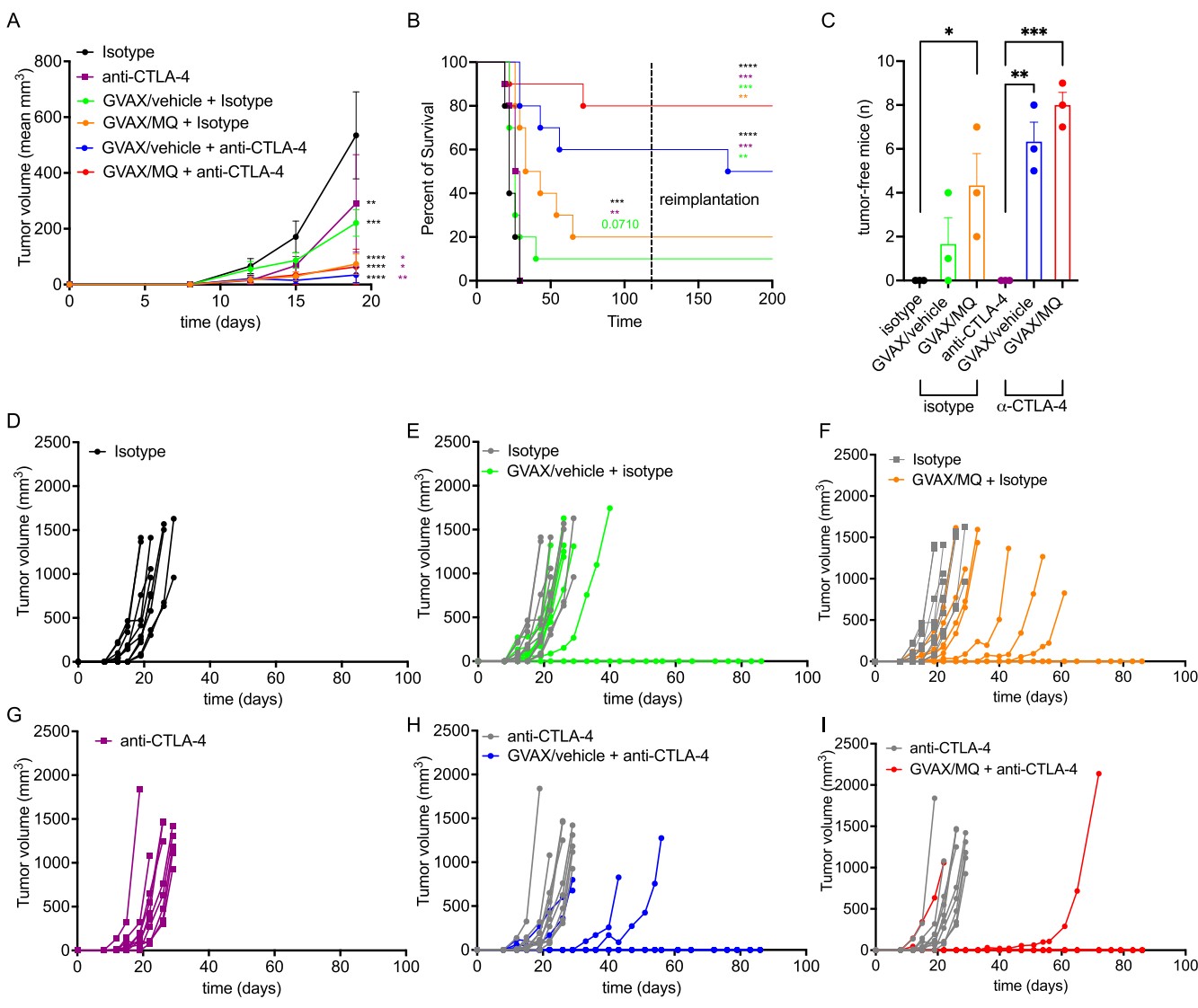

**Figure 3. The immunization effect of GVAX is enhanced with CTLA-4 blockade.**
Mice were immunized according to the schematic in Fig 2A. **(A)** Mean tumor size overtime of GVAX treatment in combination with anti-CTLA4 antibody (clone 9D9, BioXcell). An isotype antibody was used as a negative control (clone MPC11, BioXcell). **(B)** Survival of these mice, including post reimplantation with 0.1 million B16 cells at day 112. **(C)** Total number of tumor-free mice per treatment group (n = 30) when three independent experiments were pooled. **(A, B, D, E, F, G, H, I)** Individual mouse tumor volumes according to the treatment groups of the representative experiment from (A, B). The data represent mean ± SEM and the *P*-value is represented as *<0.0332, **<0.0021, ***<0.0002, ****<0.0001. *P*-value was calculated by two-way ANOVA for tumor growth curves, by log-rank (Mantel–Cox) test for survival curves and by unpaired *t* test for total number of mice comparison. n = 10, representative of three independent experiments.

once a day (qd) (Zache et al, 2008) or (ii) a novel pulse treatment in a biweekly dose of twice daily (bid) administration of APR-246 at 100 mg/kg (Fig S4A and B). Both APR-246 treatment schedules when combined with MPLA reduced tumor progression (Fig S4C, left panel) and improved overall survival (Fig S4C, right panel). As tumor immune infiltration was more pronounced using the APR-246 pulse regimen (Fig S4D) when compared with the continuous APR-246 treatment, we used the pulse regimen for subsequent studies.

A stringent test of the efficacy of a local therapy is whether it can induce systemic immunity and induce regression of distant tumors or metastases (i.e., abscopal effect). We tested whether the combination of systemic APR-246 treatment with the local treatment of

MPLA could induce regression of a distant (untreated) tumor using a bilateral flank model where B16 cells were implanted on the right and left flanks of C57BL/6 mice (Fig 5A). Whereas APR-246 treatment alone had no antitumor effect, the combination of pulsatile APR-246 treatment with MPLA reduced tumor growth (Fig 5B) of both MPLA-injected and non-injected tumors and enhancing overall survival in mice (Fig 5C). In addition, there was a concomitant increase in immune infiltration with APCs (i.e., CD11b[+] and CD11c[+] cells) and CD8[+] T cells (Fig 5D) in the MPLA-injected tumors when compared with MPLA monotherapy. Building on our previous work demonstrating the immunogenic potential of intratumoral targeting of TLR4 and CD40 (Khalil et al, 2019) with agonist agents, we

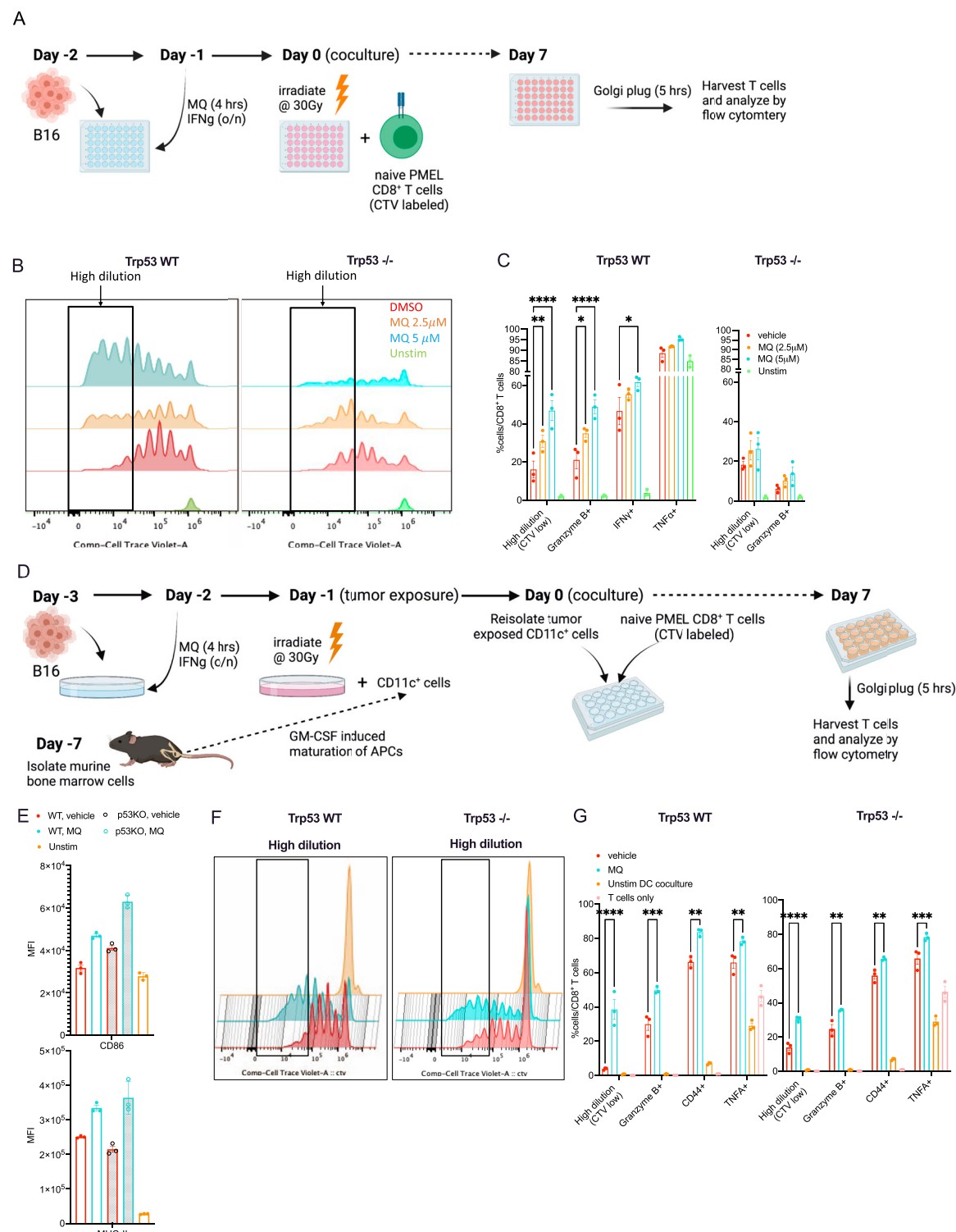

**Figure 4. MQ-treated B16 cells enhance priming and activation of antigen-specific pmel CD8⁺ T cells in vitro.**
**(A)** Schematic representation of the co-culture assay of MQ pulse-treated tumor cells with pmel CD8⁺ T cells. **(B)** Representative images of CTV dilutions in pmel CD8⁺ T cells co-cultured with B16 WT or Trp53⁻/⁻ treated cells as indicated in Fig 4A. **(C)** Quantification of the extent of CTV dilution and the relative proportions of various activation markers among these CD8⁺ T cells. **(D)** The schematic of the corresponding assay schedule using tumor-exposed dendritic cells (DC) to activate pmel CD8⁺ T cells. **(E)** The mean fluorescence intensity of CD86 (top panel) and MHC-II (bottom panel) among the reisolated tumor-exposed CD11c⁺ cells. **(F)** Representative images of the analysis of CTV dilutions of pmel CD8⁺ T cells. **(G)** The quantification of CTV dilution and the relative proportions of various activation markers among these CD8⁺ T cells when co-cultured with DCs that were exposed to B16 WT (left panel) or Trp53⁻/⁻ (right panel) cells. The data represent mean ± SEM and the *P*-value was calculated by two-way ANOVA and represented as *<0.0332, **<0.0021, ***<0.0002, ****<0.0001. n = 3 and representative of three independent experiments.

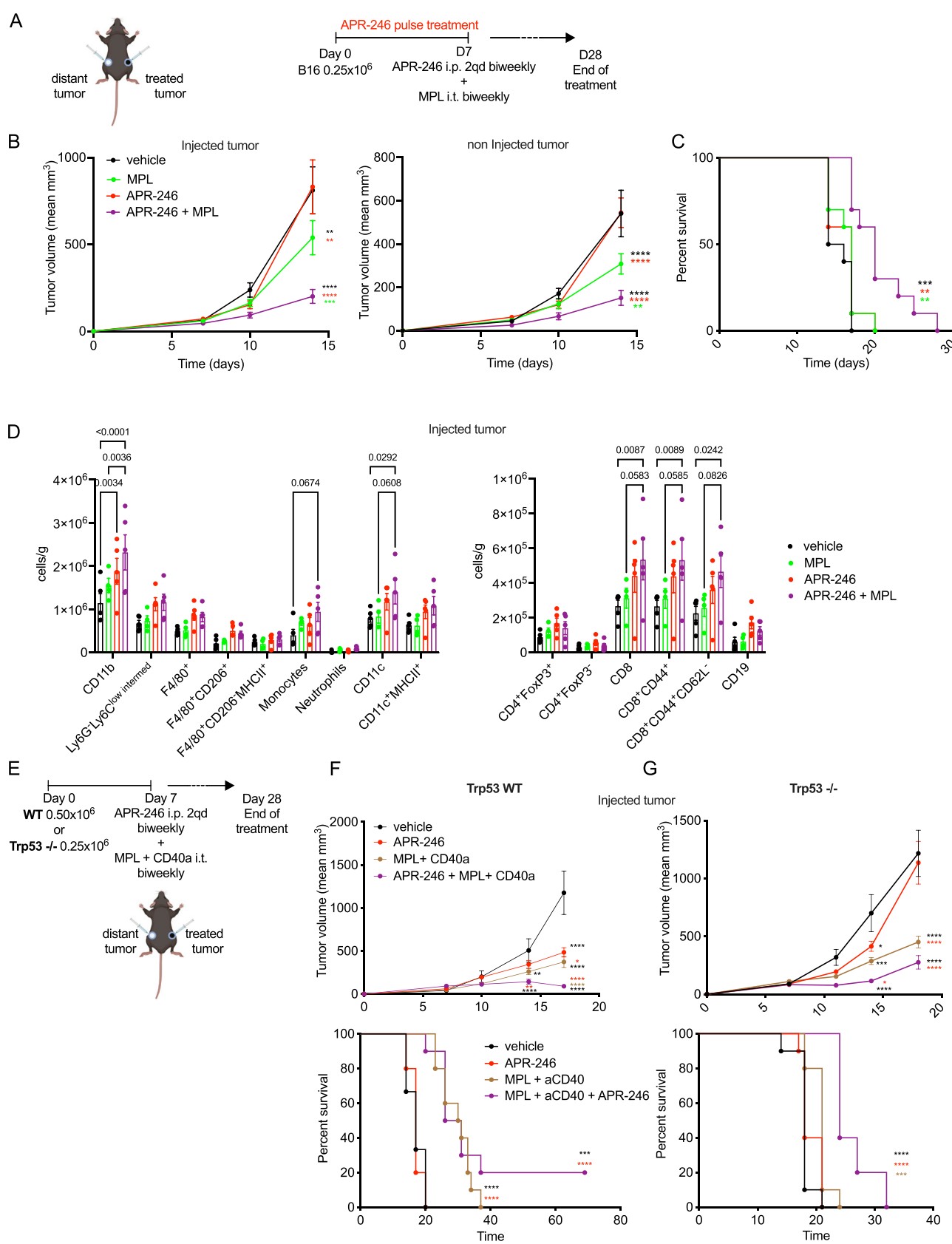

further explored the impact of adding a CD40 agonist along with the combination of MPLA and APR-246 pulse treatment. We evaluated the efficacy of this triple combination in mice bearing Trp53 WT or Trp53−/− B16 tumors (Fig 5E). We observed decreased growth of B16 WT tumors that were injected with MPLA and CD40 agonist (CD40a) (Fig 5F, upper panel) and a modest effect in the non-injected contralateral tumors (Fig S4E) for this combination. This treatment regimen also increased the overall survival of these mice (Fig 5F, lower panel). 2 out of 10 mice showed complete regression of both tumors and remained tumor free even after reimplantation of tumor cells. B16 Trp53−/− tumors are much more aggressive than their parental WT counterparts (Fig 5F and G). The combination of APR-246, MPLA, and CD40 agonist was also able to significantly reduce tumor growth and extend survival even in mice bearing these B16 tumors that lack p53 (Figs 5G and S4E and F). This underscores the efficacy of the regimen and demonstrates the relevance of the immune modifying capacity of APR-246 even in the absence of p53.

### The immunogenic effect of MQ is recapitulated with a generic cysteine binder, iodoacetamide

MQ is a Michael acceptor that can reversibly react with thiol/sulfhydryl groups (SH groups) to form thioether bonds (Lambert et al, 2009). Thiol reactivity characterizes the chemistry of cysteine, one of the least abundant amino acids, yet one that is highly concentrated and conserved in functional sites of proteins (Marino & Gladyshev, 2012; Poole, 2015). In addition to binding through one or more cysteines in the p53 core domain, MQ also reacts with other cysteine-rich and selenocysteine-rich proteins, such as thioredoxin reductase. This is the molecular basis of the reported p53-independent effects of APR-246 in cellular stress pathways[24–28].

Iodoacetamide is a broad-spectrum, irreversible cysteine-reactive compound widely used in various molecular and mass spectrometry-based applications (Nelson et al, 2008; Wiśniewski et al, 2009). To understand if the cysteine-binding property of MQ is key to its effect on tumor immunogenicity, we assessed the ability of iodoacetamide to induce similar effects. We observed a significant increase in the expression of MHC-I (H2-K$^b$) and MHC II in B16 (Fig S5A) and B16-derived Trp53−/− tumor cells when treated with iodoacetamide as compared with vehicle (Fig S5B). We also found that a pulsatile iodoacetamide treatment elicits similar immunogenic responses as a pulsatile MQ treatment. B16 cells treated with iodoacetamide were able to increase the proliferation of pmel CD8$^+$ T cells in vitro (Fig S5C and D) and to lesser degrees in vivo (Fig S5E–G). Iodoacetamide treatment also improved the antitumor effect of GVAX (Fig S5H)

with an enhanced antigen specific systemic immune response as measured by the increase (Fig S5I) and activation (i.e., CD44$^+$CD62L$^-$ and PD-1$^+$ fraction) (Fig S5J) of tetramer-positive gp100$_{25–33}$/D$^b$ reactive CD8$^+$ T cells (but not of T cells with irrelevant specificity, e.g., SIINFEKL, OT-I) in the peripheral blood of immunized mice. These results with iodoacetamide suggest that MQ may exert immunogenic effects on tumor cells through its cysteine binding ability independent of p53.

## Discussion

In this study, we evaluated the ability of APR-246 and its active constituent MQ to exert an immunogenic effect directly on tumor cells. In support of this tumor intrinsic effect, we observed increased antigen presentation in murine melanoma cells when treated with MQ/APR-246. We also demonstrated that murine tumor cells exposed to MQ have the capacity to activate and induce proliferation of tumor antigen-specific T cells in vivo and in vitro. Moreover, MQ showed the potential to enhance the immunization effect of a GVAX cell-based vaccine alone and in combination with anti-CTLA-4. The current work evaluated the impact of MQ/APR-246 on the antigenicity on the tumor cells, although the host is not treated with the p53 stabilizer. This does not preclude an impact on the p53 expression in the host by APR-246 treatment, which may further boost the effect on the tumor cells. Furthermore, we rationally designed a treatment regimen that potentiates the immunogenic effect of APR-246 in vivo when combined with intratumoral treatment using the TLR4 agonist MPLA and a CD40 agonistic monoclonal antibody. This treatment regimen was effective irrespective of the p53 status of the tumor. Our results indicate that pulsatile APR-246 treatment (biweekly, twice daily) is the ideal treatment schedule to exploit the immunogenic effect of the drug in immunotherapy combination regimens. This is based on our data on in vitro MHC-I increases in tumor cells and enhanced tumor immune infiltration in vivo upon pulsatile versus continuous APR-246 treatment regimens. Moreover, these results show that none of the various indicators of increased tumor antigenicity induced by APR-246 were entirely dependent on the presence of WT p53 as these effects were also observed in Trp53−/− tumors.

Our results support the conclusion that APR-246 exhibits antigenic and immunogenic effects on tumor cells. These effects do not appear to be directly driven by p53 but rather enhanced when p53 is present, as shown by a consistently stronger effect in the Trp53 WT tumors when compared with the Trp53−/− tumors. Thus, we hypothesize that the underlying mechanism is likely facilitated in the presence of p53 rather than being driven by the activity of p53.

**Figure 5. Combination of APR-246 and TLR4 agonist increases the tumor infiltration of antigen presenting and CD8$^+$ T cells in mice.**
**(A)** Schematic representation of the corresponding treatment schedule for the intra-peritoneal (i.p.) administration of APR-246 and the intratumoral injection (i.t.) of the TLR4 agonist, monophosphoryl lipid A (MPLA) in a bilateral flank tumor model. **(B, C)** The mean tumor size (B) and overall survival of mice (C) under the aforementioned combination treatment. Triethylamine i.t. injection was used as a vehicle control for MPLA. **(D)** The phenotype of the tumor-infiltrating myeloid and lymphoid cells under the aforementioned treatment schedule was assessed in the MPLA-injected tumors using flow cytometry. **(E)** Treatment schematic for the triple combination of i.t. MPLA and CD40 agonist (clone FGK45, BioXcell) with i.p. pulse APR-246. An isotype was used as negative control for CD40 agonist (clone 2A3, BioXcell). **(F, G)** The corresponding tumor growth and mice survival analysis of (F) WT and (G) Trp53−/− B16 tumors. The data represent mean ± SEM and the P-value is represented as *<0.0332, **<0.0021, ***<0.0002, ****<0.0001. P-value was calculated by two-way ANOVA, by log-rank (Mantel–Cox) test for survival curves. n = 10 for (B, E) and n = 5 for (D) and representative of two (D, E) to three (B) independent experiments.

We also suggest a potential mechanism that could be involved in driving this immunogenic role of APR-246 that could be driven by its ability to bind to reactive cysteines on its target proteins. MQ is a Michael acceptor, a class of small molecules that are reactive with cysteine and show high selectivity towards thiols to form a thio-ether bond. Cysteine is a relatively rare amino acid (1.7% of the residues present in eukaryotic proteins) that is concentrated in key functional sites in proteins and enzymes and in MHC-associated peptides (14% of MHC-I restricted peptides). Cysteine residues (sulfhydryl [SH] group or thiol) are the most chemically reactive of the 20 common amino acids and often undergo posttranslational modification by forming covalent disulfide bonds with other cysteines (Marino & Gladyshev, 2012; Poole, 2015; van der Reest et al, 2018; Maurais & Weerapana, 2019). The human proteome contains many cysteines that are potential ligands, revealing that covalent chemistry can be used to expand the druggable component of the human proteome (Backus et al, 2016). Posttranslational modification of cysteine residues in MHC-associated peptides decreases the binding affinity by fivefold but increases recognition by T cells by over 1,000-fold as described for the male-specific H-Y antigen that is the most extensively studied transplantation antigen (Meadows et al, 1997). The cysteine modifications of peptides occur naturally and have been studied for viral peptides and the SMCY protein of the Y chromosome acting as a minor histocompatibility antigen that evokes immune transplant rejection or graft-versus-host-disease (Meadows et al, 1997; Chen et al, 1999). In line with this, we chose to evaluate the ability of the broad-spectrum, irreversible cysteine binder iodoacetamide to also enhance the immunogenicity of B16 tumor cells. Indeed, we observed similar immunogenic effects with iodoacetamide as with MQ-treated tumor cells. Further experiments will be needed to deeply characterize the mechanism leading to tumor immunogenicity upon treatment with drugs that can behave as Michael acceptors.

The preclinical data from our current study further support the rationale from our prior preclinical work (Ghosh et al, 2022) to combine APR-246 with immunotherapy in cancer patients (Park et al, 2022) and extend the potential applicability of this combination to patients irrespective of their tumor's Trp53 status. The preliminary data from patients' samples of the clinical trial with APR-246 plus pembrolizumab showed a systemic reduction in M2-polarized myeloid cells and an increase in T cell proliferation in those patients that responded to the therapy (Ghosh et al, 2022). The results of our current study would suggest that the increase in T cell proliferation in these responders may be a combination of p53-independent and p53-dependent functions of APR-246. Thus, we believe that APR-246 would be of particular interest for the treatment of tumor types known to be highly infiltrated with immunosuppressive myeloid cells that are known to limit the efficacy of ICB (e.g., ovarian cancer and anaplastic thyroid carcinoma). The fact that APR-246 induces an immunogenic effect even in the absence of p53 in tumor cells points to a potentially broad use of APR-246 in combination with immunotherapy in the clinic.

In conclusion, in this study, we uncovered the ability of APR-246, and potentially cysteine-binding agents in general, to improve the otherwise weak immunogenicity of tumor cells leading to enhanced priming of antigen-specific CD8⁺ T cells in the TME. Our data indicate that the ability of APR-246 to bind to cysteine may be a determinant of its immunogenic effect and support the novel concept that targeting cysteine with Michael acceptors may be an effective strategy to potentiate cancer immunotherapy, including vaccines and immune-based therapies such as checkpoint blockade.

# Materials and Methods

### Cell lines, culture conditions, and chemicals

The melanoma murine cancer cell line B16F10 (referred to as B16) was obtained from ATCC. B16-GMCSF was generated by I. Hara and A. Houghton at MSKCC by retroviral transduction of B16F10 with a retroviral vector encoding the gene for mouse GM-CSF (Turk et al, 2004). Cells were maintained in RPMI medium supplemented with 10% FCS and penicillin with streptomycin (complete RPMI media). B16 Trp53 knock-out (Trp53 −/−) cells were generated by a CRISPR knockout kit (Origene KN318278) using nucleofection (Cell Line Nucleofector Kit V Catalog# VCA-1003 with Nucleofector II; Lonza) with gRNA 5′-ATAAGCCTGAAAATGTCTCC-3′ (KN318278G2) and donor vector KN318278D. B16F10 transfected with a negative scramble control vector GE100003 and the donor vector KN31827 were used as control cells (B16-WT). Clonal transformants were selected using 1 $\mu$g/ml puromycin (hydrochloride ant-pr-1 from InvivoGen). PCR confirmed the Homology Directed Repair-mediated knockout, recombination or insertion or deletion. Cell lines were routinely screened to avoid mycoplasma contamination and maintained in a humidified chamber with 5% CO2 at 37°C for up to 1 wk after thawing before injection in mice. MQ and APR-246 were provided by Aprea and iodoacetamide was obtained from Sigma-Aldrich. The complete RPMI media were prepared by the MSKCC media core. Unless otherwise indicated, other supplements for cell culture were purchased from Gibco-Invitrogen, plasticware from Corning B.V. Life Sciences, and chemicals from Sigma-Aldrich. GMCSF expression was measured in the supernatant of MQ- or vehicle-treated B16-GMCSF cells with a mouse GM-CSF Quantikine ELISA Kit (MGM00; R&D Systems).

### Mice

C57BL/6 (6–8-wk old) and Rag KO mice were purchased from Jackson Laboratory. Pmel-1 (pmel) TCR transgenic mice that have been previously reported (Overwijk et al, 2003) were bred with Thy1.1 + C57BL/6J mice (The Jackson Laboratory) as a source of tumor-specific CD8⁺ T cells. Batf3 KO mice were bred in the laboratory as previously reported (Khalil et al, 2019). All mice were maintained in micro isolator cages and treated in accordance with the NIH and American Association of Laboratory Animal Care regulations. All mouse procedures and experiments for this study were approved by the MSKCC Institutional Animal Care and Use Committee.

### In vivo reagent and treatments

APR-246 (2 mg) was dissolved in PBS and administered at 100 mg/kg intraperitoneally. Therapeutic in vivo mAbs anti-CD40 (FGK45) and

corresponding IgG isotype control (2A3), anti-CTLA-4 (clone 9D9) or the matched IgG isotype control (MPC11) were purchased from Bio X Cell. 9D9 (100 $\mu$g) and MPC11 (100 $\mu$g) were administered intra-peritoneally twice weekly. FGK45 (20 $\mu$g), MPLA (5 $\mu$g), and 2A3 (20 $\mu$g) were administered concurrently intratumorally twice weekly. MPLA (Sigma-Aldrich) was reconstituted as previously described (Stark et al, 2015). On day 0 of the experiments, tumor cells were implanted intradermally (i.d) in the flank (right if a unilateral model was used). $2.5 \times 10^5$ B16 Trp53 KO or $5 \times 10^5$ B16-WT were injected into C57BL/6J mice. PBS was used to resuspend the cells at the appropriate di-lution. Treatments were started at D7 post tumor implantation, as single agents or in combinations for 3 wk with the appropriate regimen for each drug. Tumors were measured twice a week with a caliper, and the tumor size was calculated based on an ellipsoid formula. Mice that had no visible and palpable tumors on con-secutive measurement days were considered complete regres-sions. Animals were euthanized for signs of distress or when the total tumor volume reached 2,500 mm$^3$.

### GVAX experiments

Mice were shaved and subcutaneously implanted with $5 \times 10^4$ B16 cells or B16 Trp53$^{-/-}$ cells respectively in the right flank. PBS was used to resuspend the cells at the appropriate dilution. These mice also received a simultaneous intradermal injection of a tumor cell vaccine into the left flank. The vaccine comprised of $10^6$ irradiated (16,000 rads) GM-CSF–producing cells, which were treated with MQ 10 $\mu$M for 4 h, iodoacetamide 20 $\mu$M for 30 min or the vehicle, and cultured for 48 h in fresh media before irradiation. The vaccine injections were then repeated 3 and 6 d later. To prepare a vaccine with B16 Trp53$^{-/-}$ cells, a mixture of $10^6$ B16 Trp53$^{-/-}$ cells treated with MQ, or the vehicle were mixed with $2 \times 10^5$ B16-GMCSF as indicated. Treatment with aCTLA-4 (clone 9D9; BioXcell) or the matched isotype IgG (clone MPC11; BioXcell) at 5 mg/kg was started 3 d later and performed every 3 d up to three injections. Tumor growth was scored by measuring perpendicular diameters. Mice were killed when the tumors displayed severe ulceration or reached a volume of 2,500 mm$^3$. For the memory assessment study, mice with complete responses were reimplanted with $1 \times 10^6$ B16 tumor cells (90 d after the original tumor implant).

### Adoptive transfer

gp100-specific TCR transgenic Pmel female mice (Overwijk et al, 1998, 2003) (6–12-wk-old) that had yet to begin showing signs of coat depigmentation were used as donors. CD8$^+$ T cells were pu-rified from their spleens using mouse CD8a (Ly-2) MicroBeads for positive selection (Miltenyi Biotec Inc.) and were labelled with CellTrace Violet (CTV) or CFSE (Invitrogen$^{TM}$), according to the manufacturer's suggested protocol. For adoptive transfer, the T cells were washed twice with PBS, resuspended at 0.5–1 million cells per 200 $\mu$l, and injected via tail vein into recipient animals. Recipient mice received 600 cGy total body irradiation from a $^{137}$Cs source several hours before adoptive transfer. The day after the adoptive transfer, mice were injected with a tumor cell vaccine of $10^6$ irradiated (3,000 rads) and treated B16 WT or Trp53$^{-/-}$ cells. The cells were treated in a pulse fashion with MQ 10 $\mu$M for 4 h or

iodoacetamide 20 $\mu$M for 30 min or the vehicle and cultured for 48 h in fresh media before being used for the vaccine. For B16-GMCSF–based cell vaccines, preparation was performed as per the GVAX experiment (see dedicated section above). The draining lymph nodes were harvested 5 or 6 d after the vaccination.

### NIH tetramers

H-2D(b)-restricted mouse gp100$_{25-33}$ (KVPRNQDWL) tetramer con-jugated to Alexa488 and H-2K(b)-restricted OVA$_{257-264}$ (SIINFEKL, OT-1) tetramer conjugated to PE were obtained through the NIH Tet-ramer Core Facility. Blood leukocytes were collected in heparin-coated tubes and erythrocytes were lysed using standard procedures. Remaining leukocytes were stained to detect live Tetramer$^+$CD8$^+$ cells by flow cytometry.

### Flow cytometry

Single-cell suspensions from tumors and tumor-draining lymph nodes were prepared by mechanical dissociation on 100 mM or 40 mM filters, respectively, and RBC lysis (ACK buffer, Lonza) was performed for spleens. Mouse PB was collected by retro-orbital puncture and red blood cells were lysed with Pharm Lyse Buffer (BD Biosciences). For MHC in vitro measurements, the adherent cells were collected using a non-enzymatic reagent (Cellstripper, Corning). Surface staining of mouse cells was performed after blockade of FC$\gamma$III/II receptor with purified, unlabeled CD16/CD32 antibody (clone 2.4G2; BD) and an fixable viability dye (eFluor506 or Zombie NIR), with panels of appropriately diluted fluorochrome-conjugated Abs (from BD Biosciences, eBioscience, BioLegend or Invitrogen) against the following mouse proteins in different combinations: CD45 (clone 30-F11), CD45.1 (clone A20), CD4 (clone RM4-5), CD8a (clone 5H10), Thy1.1 (clone OX-7), CD19 (clone 1D3), CD11b (clone M1/70), Ly-6C (clone AL-21), Ly-6G (clone 1A8), F4/80 (clone BM8), CD11c (clone N418), PD-1 (clone RMP1-30), CD44 (clone IM7), CD62L (clone MEL-14), CD86 (clone GL1), I-A/I-E (clone M5/114.15.2), H2Kb (clone AF6-88.5), CD107A(clone 1D4B). For intracel-lular staining, mouse cells were fixed and permeabilized (Foxp3 fixation/permeabilization buffer, eBioscience) and incubated with appropriately diluted antibodies: Foxp3 (clone FJK-16 s), Granzyme B (clone GB11), TNF-$\alpha$ (clone MP6-XT22), IFN-$\gamma$ (clone XMG1.2). For the simultaneous assessment of $\Delta\psi$m and plasma membrane integrity, cells were collected, washed, and co-stained with 40 nM 3,3'-dihexyloxacarbocyanine iodide [DiOC6(3), a $\Delta\psi$m-sensitive dye] and 1 $\mu$g/ml propidium iodide (PI) (Molecular Probes–Life Tech-nologies), following established protocols (Kepp et al, 2011). Samples were acquired and electronically compensated on a LSR II (BD), Fortessa or Aurora (Cytek) and exported for analysis in FlowJo (Tree Star, Inc.).

### Immunoblotting

For the preparation of total protein extracts, B16 cells were washed with cold PBS and lysed in a RIPA buffer containing 1% NP40, 50 mM Tris–HCl, 150 mM NaCl, 0.5% sodium deoxycholate, 0.1% sodium dodecyl sulfate containing protease and phosphatase inhibitors (Thermo Fisher Scientific). Lysates were separated on precast 4–12%

polyacrylamide NuPAGE Novex Bis-Tris gels (Invitrogen), electro transferred to nitrocellulose and probed with primary antibodies (from Cell Signaling Technology, Life Technologies Corporation, and Abcam) specific for β-actin (clone C4, Santa Cruz Biotechnology) and p53 (clone 1C12, Cell Signaling Technologies). Finally, membranes were incubated with suitable secondary IgG conjugated to horseradish peroxidase (Santa Cruz Biotechnology), followed by chemiluminescence detection with the Western Lighting Plus ECL (Perkin-Elmer) or SuperSignal West Pico reagent (Thermo Scientific-Pierce, Rockford, USA) using the Invitrogen iBright FL1000 Imaging System.

## Cell viability measurement

In vitro assessments of the pulse or continuous treatment with MQ/vehicle or the titration of the radiotherapy was performed by means of a colorimetric assay based on the reduction of the highly water-soluble tetrazolium salt WST-8 [2-(2-methoxy-4-nitrophenyl)-3-(4-nitrophenyl)-5-(2,4-disulfophenyl)-2H-tetrazolium,monosodium salt] (Cell Counting Kit-8 (CCK-8), Sigma) (which exhibits an absorbance peak around 450 nm), following conventional procedures. Absorbance at 450 nm was measured on SpectraMax i3 (Molecular Devices) and—after background subtraction—WST-8 conversion data were normalized to the readings of vehicle-treated cells included in the same test plate. The viability of cells for some of the data were alternatively measured using 1:1 dilution of the CellTiter-Glo luminescent reagent (Promega G7573) with media, which was read on a SpectraMax I3 plate reader after 10 min of shaking at room temperature. The intensity of luminescence was normalized to that of DMSO control.

## Direct priming assay

On day 1, 0.02 M/well tumor cells were plated in 48 wells in complete RPMI media. On day 2, these cells were treated with MQ or Iodoacetamide (at the indicated concentration) for 3.5 h or 30 min respectively in RPMI media lacking NEAA. The cells were then washed and treated with IFN γ (10 ng/μl) overnight in the same media. On day 3, the tumor cells are washed 2X with PBS and then irradiated at 30 Gy in 0.5 ml of media (complete RPMI media supplemented with β-mercaptoethanol). These irradiated tumor cells are then cocultured with 0.5 M/well of naïve CD8⁺ cells isolated from female pmel mice (6–8-wk-old) using mouse CD8a (Ly-2) MicroBeads for positive selection (Miltenyi Biotech Inc.) or EasySep Mouse CD8⁺ T Cell Isolation Kit (Stemcell Technologies). The ratio of tumor cells to CD8⁺ T cells at the start of co-culture was about 1:5 and the total volume of media was 1 ml per well. The CD8⁺ T cells were labelled with CTV (Invitrogen) before the co-culture. On day 7 post co-culture, the wells were treated with 1x Golgi Plug (BD Biosciences) for 5 h at 37°C, and then, the T cells were harvested by resuspension of media (the adherent cells were not collected) for analysis by flow cytometry. Multiple generations of proliferating pmel CD8⁺ T cells are plotted using FlowJo's proliferation modelling to quantify the differential dilution of the fluorescent dye CTV.

## APC co-culture assay

APCs were obtained by maturing murine bone marrow-derived cells (harvested from 6–8-wk-old naïve female mice) using GM-CSF (20 ng/μl) for 7 d (note that after 3–4 d, the GM-CSF was replenished). After 7 d of GM-CSF treatment, only the suspension cells were collected and CD11c⁺ cells were isolated using EasySep Mouse CD11c-Positive Selection Kit II (Stemcell Technologies). 3M CD11c⁺ cells were cocultured with irradiated tumor cells (treated with MQ in a pulse fashion before irradiation) for 24 h in a total of 15 ml media (complete RPMI media supplemented with β-mercaptoethanol). The ratio of tumor cells to CD11c⁺ T cells at the start of co-culture was about 3:1. The next day, both the suspension and adherent cells were collected from the tumor CD11c⁺ co-culture and the tumor-exposed CD11c⁺ cells were again isolated using the positive selection kit. These tumor-exposed CD11c⁺ cells were then cocultured with naïve CD8⁺ cells isolated from female pmel mice (6–8-wk-old) using EasySep Mouse CD8⁺ T Cell Isolation Kit (Stemcell Technologies). The ratio of CD11c⁺ to T cells at the start of co-culture was 1:1 and 0.5M of each type of cells was plated in a 24-well plate. The CD8⁺ T cells were labelled with CTV (Invitrogen) before the co-culture. The remaining tumor-exposed CD11c⁺ cells were stained and analyzed by flow cytometry. On day 7 post co-culture, the wells were treated with 1x Golgi Plug (BD Biosciences) for 5 h at 37°C and then the T cells were harvested by resuspension of media (the adherent cells were not collected) for analysis by flow cytometry.

Tumors cells were prepared as follows: 2M tumor cells were plated in 15-cm dishes in complete RPMI media. Next day, these cells were treated with MQ (10 μM) or iodoacetamide (20 μM) for 3.5 h or 30 min respectively in RPMI media lacking NEAA. The cells were then washed and treated with IFNγ (10 ng/μl) overnight in the same media. On the next day, the tumor cells are washed 2X with PBS and then irradiated at 30 Gy in 10 ml of media (complete RPMI media supplemented with β-mercaptoethanol).

## Cytokine multiplex

A small portion of the media (100–150 μl) of the cocultures (tumor-T cells or CD11c⁺ cells-T cells) was saved on day 7, before the addition of GolgiPlug. These media aliquots were briefly spun, and the supernatant was used for cytokine analysis. Cytokines were quantified using the MILLIPLEX MAP Mouse Cytokine/Chemokine Magnetic Bead 32 Plex Panel according to the manufacturer's instructions (Millipore).

## Statistical procedures

Where indicated, data were analyzed for statistical significance using Prism (GraphPad software, version 7.0) and reported as P-values. Data were analyzed with a two-tailed t test when comparing means of two independent groups (*P < 0.05, **P < 0.01, ***P < 0.001, ****P < 0.0001) and two-way ANOVA when comparing more than two groups (*<0.0332, **<0.0021, ***<0.0002, ****<0.0001). Evaluation of survival patterns in tumor-bearing mice was performed using the Kaplan–Meier method, and results were ranked according to the Mantel–Cox

log-rank test (*<0.0332, **<0.0021, ***<0.0002, ****<0.0001). Survival was defined as mice with tumors <2,500 mm$^3$. Detailed information of the statistical test and number of replicates used in each experiment are appropriately reported in the figure legends. All findings reported were reproducible and data shown are representative of at least two independent experiments, as specified, with comparable results in each experiment. The investigators were not blinded to allocation during experiments and outcome assessment.

## Supplementary Information

## Acknowledgements

J Michels has been a recipient of the Conquer Cancer ASCO Jane C Wright, MD, Endowed Young Investigator Award, was funded through Fondation Cancer Luxembourg, Fondation Nuovo-Soldati, Fondation de France, Fondation ARC. The following reagent(s) was/were obtained through the NIH Tetramer Core Facility: H-2D(b) Human Gp100$_{25-33}$ KVPRNQDWL Alexa488-labeled and H-2K(b) OT-1 SIINFEKL PE-labeled. Some graphics have been created with BioRender.com.

### Author Contributions

J Michels: conceptualization, formal analysis, investigation, and writing—original draft, review, and editing.
D Venkatesh: conceptualization, formal analysis, investigation, and writing—original draft, review, and editing.
C Liu: formal analysis and investigation.
S Budhu: formal analysis, investigation, and writing—review and editing.
H Zhong: formal analysis and investigation.
MM George: formal analysis and investigation.
D Thach: formal analysis and investigation.
Z-K Yao: formal analysis and investigation.
O Ouerfelli: formal analysis and investigation.
H Liu: methodology.
BR Stockwell: methodology.
LF Campesato: formal analysis and investigation.
D Zamarin: methodology.
R Zappasodi: methodology.
J Wolchok: conceptualization, formal analysis, supervision, funding acquisition, and writing—original draft, review, and editing.
T Merghoub: conceptualization, formal analysis, supervision, funding acquisition, and writing—original draft, review, and editing.

### Conflict of Interest Statement

All authors concur with the submission of this article and have no financial or other interests related to the submitted work. T Merghoub is a cofounder and holds an equity in IMVAQ Therapeutics. He is a consultant of Immunos Therapeutics and Pfizer. He has research support from Bristol Myers Squibb; Surface Oncology; Kyn Therapeutics; Infinity Pharmaceuticals, Inc.; Peregrine Pharmaceuticals, Inc.; Adaptive Biotechnologies; Leap Therapeutics, Inc.; and Aprea. He has patents on applications related to work on oncolytic viral therapy, alpha virus-based vaccine, neoantigen modeling, CD40, GITR, OX40, PD-1, and CTLA-4. O Ouerfelli Owns shares and is an unpaid member of the SAB, Angiogenex; has intellectual property rights with Y-Mabs Therapeutics and Jazz Pharmaceuticals; and has intellectual property rights with MSKCC and Johnson & Johnson. JD Wolchok is a consultant for Adaptive Biotech, Advaxis, Am-gen, Apricity, Array BioPharma, Ascentage Pharma, Astellas, Bayer, Beigene, Bristol Myers Squibb, Celgene, Chugai, Elucida, Eli Lilly, F Star, Genentech, Imvaq, Janssen, Kleo Pharma, Linnaeus, MedImmune, Merck, Neon Therapeutics, Ono, Polaris Pharma, Polynoma, Psioxus, Puretech, Recepta, Trieza, Sellas Life Sciences, Serametrix, Surface Oncology, and Syndax. Research support: Bristol Myers Squibb, Medimmune, Merck Pharmaceuticals, and Genentech. Equity: Potenza Therapeutics, Tizona Pharmaceuticals, Adaptive Biotechnologies, Elucida, Imvaq, Beigene, Trieza, and Linnaeus. Honorarium: Esanex. Patents: xenogeneic DNA vaccines, alphavirus replicon particles ex-pressing TRP2, MDSC assay, Newcastle disease viruses for cancer therapy, genomic signature to identify responders to ipilimumab in melanoma, engineered vaccinia viruses for cancer immunotherapy, anti-CD40 agonist mono-clonal antibody (mAb) fused to monophosphoryl lipid A (MPLA) for cancer therapy, CAR+ T cells targeting differentiation antigens as means to treat cancer,anti-PD-1 antibody, anti-CTLA-4 antibodies, and anti-GITR antibodies and methods of use thereof. S Budhu is an inventor on patent applications related to work on Lag3 and TIM3. R Zappasodi is an inventor on patent applications related to work on GITR, PD-1, and CTLA-4. R Zappasodi is a consultant for Leap Therapeutics and iTEOS Belgium SA. J Michels is a consultant for GlaxoSmithKline France, Brenus Pharma. BR Stockwell is an inventor on patents and patent applications involving small molecule drug discovery and ferroptosis; has co-founded and serves as a consultant to Inzen Therapeutics, Nevrox Limited, Exarta Therapeutics, and ProJenX, Inc.; serves as a consultant to Weatherwax Biotechnologies Corporation and Akin Gump Strauss Hauer & Feld LLP; and receives sponsored research support from Sumitomo Dainippon Pharma Oncology.

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
