## [Reviewer comments · Life Science Alliance]

Life Science Alliance

APR-246 increases tumor antigenicity independent of p53

Judith Michels, Divya Venkatesh, Cailian Liu, Sadna Budhu, Hong Zhong, Mariam M George, Daniel Thach, Zhong-Ke Yao, Ouathek Ouerfelli, Hengrui Liu, Brent R Stockwell, Luis Felipe Campesato, Dmitriy Zamarin, Roberta Zappasodi, Jedd Wolchok and Taha Merghoub

DOI: <https://doi.org/10.26508/lsa.202301999>

Corresponding author(s): Taha Merghoub (Weill Cornell Medicine)

Review Timeline:

Submission Date:	2023-02-18
Editorial Decision:	2023-02-22
Revision Received:	2023-10-10
Editorial Decision:	2023-10-11
Revision Received:	2023-10-17
Accepted:	2023-10-19

Transaction Report:

Please note that the manuscript was previously reviewed at another journal and the reports were taken into account in the decision-making process at *Life Science Alliance*. Since the original reviews are not subject to Life Science Alliance's transparent review process policy, the reports and author response cannot be published.

February 22, 2023

Re: Life Science Alliance manuscript #LSA-2023-01999-T

Prof. Taha Merghoub
Weill Cornell Medicine
New York

Dear Dr. Merghoub,

Thank you for submitting your manuscript entitled "APR-246 increases tumor antigenicity independent of p53" to Life Science Alliance. We invite you to submit a revised manuscript addressing the Reviewer comments.

Thank you for this interesting contribution to Life Science Alliance. We are looking forward to receiving your revised manuscript.

Sincerely,

- A letter addressing the reviewers' comments point by point.
- An editable version of the final text (.DOC or .DOCX) is needed for copyediting (no PDFs).
- High-resolution figure, supplementary figure and video files uploaded as individual files: See our detailed guidelines for preparing your production-ready images, <https://www.life-science-alliance.org/authors>
- Summary blurb (enter in submission system): A short text summarizing in a single sentence the study (max. 200 characters including spaces). This text is used in conjunction with the titles of papers, hence should be informative and complementary to the title and running title. It should describe the context and significance of the findings for a general readership; it should be written in the present tense and refer to the work in the third person. Author names should not be mentioned.
- By submitting a revision, you attest that you are aware of our payment policies found here: <https://www.life->

science-alliance.org/copyright-license-fee

B. MANUSCRIPT ORGANIZATION AND FORMATTING:

*****IMPORTANT:** It is Life Science Alliance policy that if requested, original data images must be made available. Failure to provide original images upon request will result in unavoidable delays in publication. Please ensure that you have access to all original microscopy and blot data images before submitting your revision. *******

October 11, 2023

RE: Life Science Alliance Manuscript #LSA-2023-01999-TR

Prof. Taha Merghoub
Weill Cornell Medicine
413 East 69th St., BRB-1402
New York, NY 10021

Dear Dr. Merghoub,

Thank you for submitting your revised manuscript entitled "APR-246 increases tumor antigenicity independent of p53". We would be happy to publish your paper in Life Science Alliance pending final revisions necessary to meet our formatting guidelines.

- please upload your main and supplementary figures as single files
- please add ORCID ID for the corresponding (and secondary corresponding) author--you should have received instructions on how to do so
- please add a Summary Blurb/Alternate Abstract to our system
- please add the Twitter handle of your host institute/organization as well as your own or/and one of the authors in our system
- please move your table and figure legends after the references section
- please use the [10 author names et al.] format in your references (i.e., limit the author names to the first 10)
- please add callouts for Figures 2A, C, D; S1E, F; S3C to your main manuscript text
- please rename the Disclosures section to Conflict of Interest Statement

To upload the final version of your manuscript, please log in to your account: <https://lsa.msubmit.net/cgi-bin/main.plex>

A. FINAL FILES:

-- Summary blurb (enter in submission system): A short text summarizing in a single sentence the study (max. 200 characters including spaces). This text is used in conjunction with the titles of papers, hence should be informative

and complementary to the title. It should describe the context and significance of the findings for a general readership; it should be written in the present tense and refer to the work in the third person. Author names should not be mentioned.

B. MANUSCRIPT ORGANIZATION AND FORMATTING:

Sincerely,

October 19, 2023

RE: Life Science Alliance Manuscript #LSA-2023-01999-TRR

Prof. Taha Merghoub
Weill Cornell Medicine
413 East 69th St., BRB-1402
New York, NY 10021

Dear Dr. Merghoub,

Thank you for submitting your Research Article entitled "APR-246 increases tumor antigenicity independent of p53". It is a pleasure to let you know that your manuscript is now accepted for publication in Life Science Alliance. Congratulations on this interesting work.

DISTRIBUTION OF MATERIALS:

Again, congratulations on a very nice paper. I hope you found the review process to be constructive and are pleased with how the manuscript was handled editorially. We look forward to future exciting submissions from your lab.

Sincerely,
